# Criterion-Related Validity of Field-Based Fitness Tests in Adults: A Systematic Review

**DOI:** 10.3390/jcm10163743

**Published:** 2021-08-23

**Authors:** Jose Castro-Piñero, Nuria Marin-Jimenez, Jorge R. Fernandez-Santos, Fatima Martin-Acosta, Victor Segura-Jimenez, Rocio Izquierdo-Gomez, Jonatan R. Ruiz, Magdalena Cuenca-Garcia

**Affiliations:** 1GALENO Research Group, Department of Physical Education, Faculty of Education Sciences, University of Cádiz, Avenida República Saharaui s/n, Puerto Real, 11519 Cádiz, Spain; jose.castro@uca.es (J.C.-P.); jorgedelrosario.fernandez@uca.es (J.R.F.-S.); fatima.martin@uca.es (F.M.-A.); victor.segura@uca.es (V.S.-J.); rocio.izquierdo@uca.es (R.I.-G.); magdalena.cuenca@uca.es (M.C.-G.); 2Instituto de Investigación e Innovación Biomédica de Cádiz (INiBICA), 11009 Cádiz, Spain; 3PROmoting FITness and Health through Physical Activity Research Group (PROFITH), Sport and Health University Research Institute (iMUDS), Department of Physical and Sports Education, School of Sports Science, University of Granada, 18007 Granada, Spain; ruizj@ugr.es

**Keywords:** cardiorespiratory fitness, muscular strength, motor fitness and flexibility, validation, fitness testing, adulthood

## Abstract

We comprehensively assessed the criterion-related validity of existing field-based fitness tests used to indicate adult health (19–64 years, with no known pathologies). The medical electronic databases MEDLINE (via PubMed) and Web of Science (all databases) were screened for studies published up to July 2020. Each original study’s methodological quality was classified as high, low and very low, according to the number of participants, the description of the study population, statistical analysis and systematic reviews which were appraised via the AMSTAR rating scale. Three evidence levels were constructed (strong, moderate and limited evidence) according to the number of studies and the consistency of the findings. We identified 101 original studies (50 of high quality) and five systematic reviews examining the criterion-related validity of field-based fitness tests in adults. Strong evidence indicated that the 20 m shuttle run, 1.5-mile, 12 min run/walk, YMCA step, 2 km walk and 6 min walk test are valid for estimating cardiorespiratory fitness; the handgrip strength test is valid for assessing hand maximal isometric strength; and the Biering–Sørensen test to evaluate the endurance strength of hip and back muscles; however, the sit-and reach test, and its different versions, and the toe-to-touch test are not valid for assessing hamstring and lower back flexibility. We found moderate evidence supporting that the 20 m square shuttle run test is a valid test for estimating cardiorespiratory fitness. Other field-based fitness tests presented limited evidence, mainly due to few studies. We developed an evidence-based proposal of the most valid field-based fitness tests in healthy adults aged 19–64 years old.

## 1. Introduction

Physical fitness is an integrated measure of all the functions and structures involved in performing physical activity [1]. Nowadays, physical fitness is one surrogate marker of overall adult health (19–64 years), especially cardiorespiratory fitness and muscular strength. Cardiorespiratory fitness is inversely associated with cardiovascular diseases [2], obesity [3], osteoporosis [4] diabetes [5], different cancer types [6,7], and is a predictor of all-cause of mortality [8,9,10,11,12] and cardiovascular disease [10,12,13,14,15]. Likewise, in the psychological sphere, high levels of cardiorespiratory fitness are associated with well-being [16,17], improved cognitive function [18] and a reduced risk of Alzheimer’s disease [19] and other mental conditions such as anxiety, panic and depression [20]. Muscular strength demonstrates a protective effect against all-cause mortality [21,22]; and is inversely associated with weight gain and adiposity-related hypertension occurrence and the prevalence and incidence of the metabolic syndrome, [22] and mental health clinical presentations [23,24]. Consequently, physical fitness assessment is a vital tool of prevention and health diagnoses.

Laboratory testing is an objective and accurate method of assessing physical fitness. However, due to the cost of sophisticated instruments, time constraints and the need for qualified technicians, laboratory testing is limited to sport clubs, schools, population-based studies, and offices or clinical settings. However, field-based fitness testing can offer useful and practical alternatives as screening tools, since they are relatively safe and time-efficient, involve minimal equipment and low cost, and can be easily administered to multiple people simultaneously.

The validity of field-based fitness tests needs to be considered when deciding which test to use [25]. Criterion-related validity refers to the extent to which a field-based test of a physical fitness component correlates with the criterion measure (i.e., the gold standard) [26]. Since the early interest in physical fitness testing in the 1950–1960s, many field-based fitness tests have been proposed [27]. It would be desirable to summarise the criterion-related validity of the existing field-based fitness tests in adults. There have been attempts to summarize the criterion-related validity of a certain test [28,29] or several tests with a common characteristic [30,31,32]; however, no attempts have been made to summarise the criterion-related validity of all the existing field-based fitness tests in adults. 

Therefore, the aim of the present systematic review was to comprehensively study the criterion-related validity of the existing field-based fitness tests used in adults. The findings of this review will provide an evidence-based proposal for most valid field-based fitness tests for healthy adults, aged 19–64 years old.

## 2. Materials and Methods

The review was registered in PROSPERO (registration number: CRD42019118482) and the applied methodology followed the guidelines drawn in the Preferred Reporting Items for Systematic Reviews and Meta-Analysis (PRISMA) statement [33].

### 2.1. Literature Search

The search was performed in the MEDLINE (via Pubmed) and Web of Science electronic databases from inception until July 2020. We screened studies conducted for criterion-related validity in adults, where one or more field-based fitness tests were carried out. Thus, the keywords selected were based on terms related to “criterion-related validity”, “adults” and “field-based fitness test”. The search syntax was adapted to the indexing terms of each database (see Appendix A). Searching was restricted to articles published in humans and the English or Spanish languages.

### 2.2. Eligibility Criteria

The inclusion criteria for this systematic review were the (1) age criterion: adults (19–64 years old). During this review, we faced the problem that some studies sampled adults and older adults, or adults or adolescents together. In these cases, we observed whether these studies performed stratified analyses by age groups, isolating the adult population from the rest; if so, the study was included and information concerning the adult population reported. In contrast, when the authors analysed the whole sample together, we only included the study if the age of the sample was predominantly within our study age range; (2) participants: the study population was based on a generally healthy population, who did not present any injury, physical and/or mental disabilities, irrespective of body mass index (BMI), diabetes or other cardiovascular risks (i.e., hypertension, hypercholesterolemia, lipid profiles, glucose levels, insulin sensitivity); and (3) study design: original studies or systematic reviews/meta-analysis. The original studies that were selected for the analysis of their criterion-related validity but which were also included in the selected systematic reviews were excluded; (4) language criterion: articles were only published in English or Spanish; (5) topic criterion: studies examining the criterion-related validity of the field-based fitness test. Studies examining the relationship between field-based fitness tests were excluded. Likewise, studies that analysed the criterion-related validity of tests designed for exclusive use in sports or clinical settings were not included.

Two authors (J.C.P. and J.R.F.S.) independently assessed the titles and abstracts of the articles retrieved by the search strategy for eligibility. Then, the full texts of the selected articles were acquired, and the same two researchers independently screened them to determine whether to include the article based on the inclusion criteria. When no consensus was reached between both researchers, a third research (N.M.J.) made the final decision with regard to inclusion. Reasons for the exclusion of identified articles were recorded.

### 2.3. Data Extraction

Two researchers (N.M.J. and F.M.A.) independently extracted the following information from each eligible original study according to the standardized form: (1) the author’s name; (2) participants (sex and number); (3) age of participants; (4) filed-based test; (5) criterion measure (gold standard); (6) statistical methods; (7) main outcome; and (8) conclusions. 

The same researchers independently extracted the following information from the systematic reviews: (1) author´s name, date and years covered by the review; (2) type of review and number of included studies; (3) age of participants; (4) filed-based test; (5) criterion measure (gold standard); (6) main outcome; and (7) conclusions. 

Disagreements in the extracted data were discussed between studies until a consensus was reached.

### 2.4. Criteria for Risk of Bias Assessment

Due to the heterogeneity of statistical methods employed by the original studies selected, the high number of tests included, and the limited number of studies per test, a meta-analysis was not conducted. An assessment of risk of bias in selected original studies and systematic reviews was made for each eligible study by two studies (N.M.J. and F.M.A.) independently. Discrepancies were solved in a consensus meeting. Inter-rater agreement for the risk of bias between researchers was calculated by the percentage agreement (96% (Kappa = 0.962) before consensus, and 100% agreement after consensus meeting). 

The assessing risk of bias criteria in original studies were determined according to quality assessment list employed by Castro-Piñero et al. [27], which include the three following criteria: (1) the adequate number of participants; (2) an adequate description of the study population; and (3) adequate statistical analysis (see Appendix A). Each criterion was rated from 0 to 2, being 2 the best score. For all studies, a total score was calculated by counting up the number of positive items (a total score between 0 and 6). Studies were categorized as very low quality (0–2), low quality (3–4) and high quality (5–6).

The methodological quality of each systematic review was appraised using the ‘Assessment of Multiple Systematic Reviews’ (AMSTAR) rating scale [34]. AMSTAR contains 11-items to assess the methodological aspects of reviews with items scored as 1 if the answer was “Yes”, and 0 if the answer was “No”, “Cannot Answer” or “Not Applicable” (see Appendix A). The total score ranged from 0 to 11. The item on conflict of interest requires that the systematic review and all primary studies be assessed. We modified this item to only assess the review itself as Biddle et al. [35] proposed, given that PRISMA does not require a conflict-of-interest assessment for each primary study. The final quality rates were computed by tertiles, where the first tertile ranged from 0 to 3 points (low quality); the second tertile from 4 to 7 points (medium quality); and the third tertile from 8 to 11 points (high quality).

### 2.5. Levels of Evidence

Three evidence levels [27] were constructed: (1) strong evidence: consistent findings in three or more high-quality studies; (2) moderate evidence: consistent findings in two high-quality studies; and (3) limited evidence: consistent findings in multiple low-quality studies, inconsistent results found in multiple high-quality studies, or results based on one single study. The degree of criterion-related validity of the field-based fitness test will be discussed for those tests on which we found strong or moderate evidence that the test is (or not) valid. The results of low- or very low-quality studies can be seen in the Appendix A.

## 3. Results

The literature search yielded 9202 and 27 additional records were identified through other sources (see the PRISMA flowchart in Figure 1). After the removal of duplicate references (1805 studies), and the screening of titles and abstracts (7233 studies), we excluded 9038 studies. A total of ▣191 full-text studies were assessed for eligibility, and 85 studies (six systematic reviews) were excluded due to reasons indicated in Figure 1. 

Finally, a total of 101 original studies (see Appendix A) addressed the criterion-related validity of field-based fitness tests in adults aged 19–64 years. The sample size involved 10,632 participants (see Appendix A). Eighty-six and seventy-eight original studies reported female (*n* = 5539) and male (*n* = 4722) sample proportions, respectively; however, in 7 seven studies, sex was not specified.

A total of four meta-analyses [28,29,30,31] and one systematic review [32] were included in the present systematic review (see Appendix A). The sample size involved 9985 participants with ages ranging from 19 to 64 years (see Appendix A). 

### 3.1. Quality Assessment 

Of the 101 original studies included in the present systematic review, 11 and 40 studies were classified as very low (a total score less than 2) and low quality (a total score of 3 and 4), respectively (see Appendix A). A total of 50 original studies were classified as high-quality (a total score higher than 4). Of these 40, nine and one analysed the criterion-related validity of cardiorespiratory fitness, muscular strength and flexibility field-based fitness tests in adults, respectively. No study of those classified as high quality analysed the criterion-related validity of motor fitness (i.e., speed, agility, balance and coordination).

Two meta-analyses [28,30] and one systematic review [32] were ranked as high quality (all eight points), and two meta-analyses [29,31] were ranked as medium quality (both seven points) (see Appendix A). Three of them assessed the criterion-related validity of field-based cardiorespiratory fitness tests: the 20 m shuttle run test [28]; distance and time-based run/walk tests [30]; and the step tests [32]—whilst and two of them studied the criterion-related validity of the sit-and-reach [31] and toe-to-touch tests [29].

* References of high-quality studies are presented in Appendix A.

### 3.2. Criterion-Related Validity

Table 1 shows a summary of the different levels of evidence found for the criterion-related validity of cardiorespiratory fitness tests.

#### 3.2.1. Cardiorespiratory Fitness

##### Distance and Time-Based Run/Walk Tests

Seventeen high-quality studies examined the criterion-related validity of the distance run/walk or walk tests (see Appendix A). Four and two studies showed that the 2 km walk [36,37,38,39] and 1.5-mile run/walk [40,41] tests, respectively, were valid for assessing cardiorespiratory fitness (r = 0.80–0.93, all *p* < 0.05). Four studies [42,43,44,45] observed that the 1-mile walk test was an accurate test for estimating VO_2max_ (r = 0.81–0.88, all *p* < 0.05), while another two studies [46,47] showed that it was not a valid test (r = 0.69, 13.3% E, *p* < 0.05; mean differences range from 2.360 to 9.131 mL/kg/min, all *p* < 0.001, respectively). The treadmill jogging test reported contradictory results: one study [48] found it to have high validity for assessing cardiorespiratory fitness (r = 0.84, both *p* < 0.001); whereas another study [41] revealed that it was not a valid test (r = 0.50, *p* < 0.05).

Five high-quality studies investigated the criterion-related validity of the time-based run/walk or walk tests (see Appendix A). These studies showed that the 3 min walk, [49] 6 min walk, [50,51,52] and the 12 min run/walk [41] tests were valid for assessing cardiorespiratory fitness (r = 0.70–0.95, all *p* < 0.05). Additionally, one original high-quality study reported that the University Montreal test [53] was valid for estimating cardiorespiratory fitness (r = 0.71, *p* < 0.001; mean difference = 0.025 ± 7.445 mL/kg/min., *p* > 0.05).

A meta-analysis [30] consisting of 102 studies on adults determined that the criterion-related validity of the distance run/walk field tests for estimating cardiorespiratory fitness ranged from low to high, with the 1.5-mile (r_p_ = 0.80; 95% CI: 0.72–0.80) and 12 min run/walk tests (r_p_ = 0.79; 95% CI: 0.71–0.87) being the best predictors (see Appendix A).

##### Twenty-Metre Shuttle Run Test

Nine high-quality studies analysed the criterion-related validity of the 20 m shuttle run test [41,54,55,56,57,58] or modifications of it [55,57,59,60,61] (see Appendix A). Four studies [41,55,56,57] reported that the 20 m shuttle run was a valid test for assessing cardiorespiratory fitness (r = 0.82–0.94, all *p* < 0.05). However, one study [58] concluded that this test was not valid for assessing cardiorespiratory fitness (mean differences range from −0.54 ± 6.23 to −2.94 ± 6.55 mL/kg/min, all *p* < 0.01). Two studies [59,60] proved that the incremental shuttle walk test was not valid (r = 0.72, 19% E, both *p* < 0.001), while one study [61] found that this test was valid for assessing cardiorespiratory fitness (mean difference = 0.14 ± 9.27mL/kg/min, *p* > 0.05). Moreover, two studies [55,57] reported that the 20 m square shuttle run test was valid (r = 0.95, both *p* < 0.001).

A meta-analysis [28] which included 24 studies in adults found that the 20 m shuttle run test had a moderate-to-high criterion-related validity for estimating VO_2max_ (r_p_ = 0.79–0.94; 95% CI: 0.56–1.00) (see Appendix A).

##### Step Tests

Eleven high-quality studies analysed the criterion-related validity of the step tests (see Appendix A). Four studies observed that the Danish step [62], the Queen’s College step [63], and the 2 min step [64] tests were not valid for estimating VO_2max_ (r = 0.034–0.72, all *p* < 0.05). However, another eight studies proved the validity of the modified Canadian aerobic fitness [65], 6 min single 15 cm step [66], YMCA step [67,68,69,70,71], Tecumseh step [70] and modified Harvard step [72] tests (r = 0.80–0.91, all *p* < 0.05).

A systematic review [32] comprised of 11 studies on adults investigated the criterion-related validity of the step tests (see Appendix A). Validity measures were varied, and a broad range of correlation coefficients were reported across the 11 studies (r = 0.469–0.95; all *p* < 0.005) with conflicting results in most of the step test protocols. The study concluded that the Chester step test was the best predictor for assessing cardiorespiratory fitness.

#### 3.2.2. Muscular Strength

Table 2 shows a summary of the different levels of evidence found for the criterion-related validity of muscular strength, flexibility and motor fitness tests.

##### Maximal Isometric Strength

Four high-quality studies assessed the criterion-related validity of hand maximal isometric strength, using the handgrip strength tests (see Appendix A). Three high-quality studies reported that the TKK dynamometer [73,74,75] was valid (mean difference range −0.20, *p* > 0.05 to 2.02 kg *p* < 0.001) (r = 0.98, *p* < 0.001). However, three studies showed inconclusive results about the validity of the DynEx dynamometer [73,75,76], and two studies observed that the Jamar dynamometer [73,76] was less accurate than the TKK and DynEx dynamometer for estimating hand maximal isometric strength.

##### Endurance Strength

Four high-quality studies assessed the criterion-related validity of trunk endurance strength (see Appendix A). Two studies [77,78] suggested that the Biering–Sørensen (r = 0.84–98, *p* < 0.01) test was valid, whereas another study [79] reported acceptable validity (r = 0.60–0.71, *p* < 0.05). One study showed that the prone bridging test [80] was valid for assessing trunk endurance strength (no mean difference, *p* > 0.05).

##### Explosive Strength

Only one high-quality study assessed the criterion-related validity of explosive strength (see Appendix A). This study concluded that the Sargent test [81] was not valid (mean difference: 4.4 ± 5.1, *p* < 0.001) for estimating lower body explosive strength.

#### 3.2.3. Flexibility

Only one study [82] that examined the criterion-related validity of flexibility tests was classified as high quality (see Appendix A). They found that the sit-and-reach was not a valid test (r = 0.44–0.48, *p* < 0.05). 

A meta-analysis [31] which included 28 studies on adults (see Appendix A) found that the sit-and-reach test and its different versions, had moderate validity for estimating hamstring extensibility (r_p_ ranged from 0.49; 95% CI: 0.29–0.68 to 0.68; 95% CI: 0.55–0.80), but a low validity for estimating lumbar extensibility (r_p_ ranged from 0.16; 95% CI: −0.10–0.41 to 0.35; 95% CI: 0.15–0.54). Moreover, another meta-analysis [29] carried out on adults (of six studies) reported that the toe-touch test had moderate validity for assessing hamstring extensibility (r_p_ = 0.66; 95% CI: 0.56–1.00).

#### 3.2.4. Motor Fitness

No study investigating the criterion-related validity of motor fitness tests was classified as high quality (see Appendix A).

## 4. Discussion

The present systematic review comprehensively studied the criterion-related validity of the existing field-based fitness tests used in adults. The findings of this review provide an evidence-based proposal for most valid field-based fitness tests for adult population.

### 4.1. Cardiorespiratory Fitness

The gold standard to assess VO_2max_ is the Douglas bag method, although there is agreement that the respiratory gas analyser is a valid method of assessing oxygen uptake [83]. All high-quality studies measured VO_2max_ or peak oxygen consumption when performing a submaximal/maximal treadmill or cycle test, except Manttari et al. [52], who directly measured VO_2max_ when performing the 6 min walk test.

#### 4.1.1. Distance and Time-Based Run/Walk Tests

The run/walk field tests are probably the most widely used tests [27,84], however, until recently, there was no consensus regarding the most appropriate distance or time to use for these tests [85]. Mayorga et al. [30] performed a meta-analysis which examined the criterion-related validity of the 5000 m, 3 mile, 2 mile, 3000 m, 1.5-mile, 1-mile, 1000 m, ½-mile, 600 m, 600 yd, ¼-mile, 15 min, 12 min, 9 min, and 6 min run/walk tests. They found that the criterion-related validity of the run/walk tests, only considering the performance score, ranged from low to high, with the 1.5-mile and the 12 min run/walk tests being the most appropriate tests for estimating cardiorespiratory fitness in adults aged 19–64 years. Sex, age or VO_2max_ level did not affect criterion-related validity, whereas when multiple predictors (i.e., performance score, sex, age or body mass) were considered, the criterion-related validity values were higher. In this sense, two high-quality original studies reinforced these results, and showed that the 12 min [41] and the 1.5-mile [40,41] run/walk tests were fairly accurate for estimating cardiorespiratory fitness in adults aged 18–26 years (r = 0.87–0.93, *p* < 0.05).

Overall, the run/walk tests are not user-friendly tests, due to the difficulty of developing an appropriate pace, which may affect the test outcome (some participants start too fast, so they are unable to maintain their speed throughout the test; others start too slow, so when they wish to increase their speed the test is already finished). These problems are more likely to occur in longer distance tests. Other factors affecting the test outcome include the individual’s willingness to endure the discomfort of strenuous exercise, a short attention span, poor motivation, and limited interest in a monotonous task [86,87,88].

The 2 km and 6 min walk tests are probably the most widely used walk tests in adults [39,51]. Both tests require submaximal effort, thus avoiding the problem of enduring the discomfort of strenuous exercise. In addition, it allows to evaluate those people with a low level of physical fitness or is unable to run. Three high-quality studies [36,37,39] observed that Oja’s equation derived from the 2 km walk test has high validity (r = 0.80–0.87, all *p* < 0.05) in untrained and/or overweight/obese adults aged 20–64 years. One high-quality study reported that the 2 km walk test [38] is a reasonably valid field test for estimating the cardiorespiratory fitness of moderately active adults aged 35–45 years, but not in adults with very high maximal aerobic power. 

Many studies developed prediction equations for the 6 min test based on spirometry [89]. However, only three high-quality studies [50,51,52] analysed the criterion-related validity of the 6 min test based on VO_2max_ in adults. They showed a moderate-to-high validity (r = 0.70–0.93, all *p* < 0.001) in obese and healthy adults aged 18–64 years. Burr et al. [90] suggested that, on its own, the 6 min walk test can be useful to discriminate between broad categories of high, moderate and low fitness, but that this approach may be associated with a degree of error, especially in the high fitness group.

According to these findings, the 2 km and 6 min walk tests are valid for use in adults aged 19–64 years with low or moderate fitness levels, but not in adults with a high fitness level.

Regarding the 1 mile walk test, conflicting results were found, especially when examining the accuracy of the Kline’s [42] and Dolgener’s [46] equations in adults aged 19–64 years.

#### 4.1.2. Twenty-Metre Shuttle Run Test

The 20 m shuttle run test was developed by Leger at al. [91] to solve the pace issue of the run/walk tests. The test consists of 1 min stages of continuous running at an increasing speed. Recently, a meta-analysis [28] showed that the performance score of the 20 m shuttle run test had a moderate-to-high criterion-related validity for estimating VO_2max_ (*r*_p_ = 0.66–0.84) in youth and adults aged 18–64 years, higher than when other variables (i.e., sex, age or body mass) were accounted for (*r*_p_ = 0.78–0.95). This study also reported that Leger’s protocol had a greater average criterion-related validity coefficient (*r*_p_ = 0.84; 95% CI: 0.80–0.89) than Eurofit, QUB and Dong-HO protocols; and Leger’s protocol was statistically higher for adults (*r*_p_= 0.94, 0.87–1.00) than for children (*r*_p_ = 0.78; 95% CI: 0.72–0.85). These values are higher than those reported for the 1500 m and 12 min run/walk tests [30]. Moreover, the meta-analysis showed that sex did not seem to affect the criterion-related validity values.

On the other hand, Cooper et al. [54] showed that Brewer’s protocol and equation were not valid for assessing active young people aged 18–26 years (mean difference = 1.8 ± 6.3 mL/kg/min; *p* = 0.004). In line with these findings, Kim et al. [58] observed that Leger’s protocol and equation were more accurate than Brewer’s protocol and equation (mean difference −0.54 mL/kg/min; %CV: 1.39 vs. mean difference −2.944 mL/kg/min; %CV: 8.87) in Korean adults, especially in women. Nonetheless, the authors suggested the need to develop new equations for Korean adults.

It is important to note that the 20 m square shuttle run test [55,57] was proposed as an alternative to the 20 m shuttle run test to reduce the test’s turning angle from 180 to 90. This test was the best predictor of VO_2max_ than the 20 m shuttle run test in young male adults aged 18–25 years.

#### 4.1.3. Step Tests

Step tests are a safe, simple, inexpensive and practical method of assessing cardiorespiratory fitness under submaximal conditions, which require minimum space [32]; they are also a great alternative to laboratory tests in clinical settings. There are a wide variety of step test protocols which differ in terms of stepping frequency, test duration and number of test stages. Bennett et al. [32] analysed the criterion-related validity of different step tests (the Chester step test, a personalised step test, the STEP tool step test, the Queen’s College step test, the Skubic and Hodgkins step test, a height-adjusted, rate-specific, single-state step test, the Astrand–Ryhming step test, and a modified YMCA 3 min step test) in adults aged 18–64 years. The validity of these tests ranged from moderate to high, and they suggested that the Chester step test was the most valid step test to evaluate cardiorespiratory fitness in adults. However, this systematic review only included two studies with contradictory results, similarly to the Queen’s College step test.

Analysing the 12 high-quality studies that examined the criterion-related validity of the step tests in adults aged 19–64 years, we can conclude that the YMCA step test [67,71] seemed to be the most appropriate step test to estimate VO_2max_ in adults aged 19–64 years. However, it is important to note that there is no single equation, since the result of the equation depends on the sample used. Santo and Golding [92] even altered the protocol by adjusting the step height to the individual participant’s height in order to increase the accuracy of this test.

#### 4.1.4. Levels of Evidence

Strong evidence indicated that (a) the 20 m shuttle run test using Leger’s equation, the 2 km walk using Oja’s equation, the 6 min and the YMCA step tests are valid for estimating cardiorespiratory fitness; and (b) the criterion-related validity of the distance and time-based run/walk tests range from low to high, with the 1.5-mile and 12 min run/walk tests being the best predictors. Moderate evidence indicated that the 20 m square shuttle run test is valid for estimating cardiorespiratory fitness. Due to the inconsistent results found in high-quality studies, limited evidence was found for the validity of the 1-mile walk, treadmill jogging, incremental shuttle walking, Chester, and Queen’s College step tests. Due to the low number of high-quality studies, limited evidence indicated that (a) the 3 min walk, the ¼-mile walk, Mankato submaximal, modified Astrand–Ryhming, University Montreal, modified Canadian aerobic fitness step, 6 min single 15 cm step, Tecumseh step, modified Harvard step and Astrand–Ryhming Step tests are valid for estimating cardiorespiratory fitness; and (b) the YMCA cycle, Ruffier, Danish step, and 2 min step tests are not valid for estimating cardiorespiratory fitness. Due to the consistent results found in multiple low-quality studies, limited evidence supported using the 6 min step test for estimating cardiorespiratory fitness.

### 4.2. Muscular Strength

The specificity of the type of muscular work performed and the use of different energy systems are both major challenges for establishing a gold standard method for maximal, endurance and explosive muscular strength tests [93]. One repetition maximum (1RM) and repetitions to a certain percentage of 1RM (i.e., 50% of 1RM or 70% of 1RM) [27], isokinetic dynamometer strength [94,95,96], and electromyography [78,80] were used as gold standards.

#### 4.2.1. Maximal Isometric Strength

The TKK dynamometer [73,74,75] seemed the most appropriate test to assess maximal isometric strength in adults. All the studies used the “known weights” as the criterion reference. 

Several studies examined whether the elbow position (extended or flexed at 90 degrees) affected the hand maximal isometric strength score in children [75], adolescents [97] and young adults [98]. They observed that performing the handgrip strength test with the elbow extended seems the most appropriate protocol to evaluate hand maximal isometric strength in these populations—which is in accordance with the protocol recommended by the American Center for Disease Control and Prevention [99].

Ruiz et al. [100] also investigated whether the position (grip span) on the standard grip dynamometer determined the hand maximal isometric strength in adults. They found that when measuring hand maximal isometric strength in women, hand size must be taken into consideration, providing the mathematical equation (*y* = *x*/5 + 1.5 cm) to adapt optimal grip span (*y*) to hand size (*x*). In adult men, optimal grip span could be set at a fixed value (5.5 cm) and is not influenced by hand size.

Importantly, just like the step test, the handgrip strength test can be very useful in clinical settings because it requires minimal equipment and space, is time-efficient and easy to administer.

#### 4.2.2. Endurance Strength

The Biering–Sørensen test, a trunk holding test in an antigravity prone position, is commonly used to measure the back and hip muscle endurance strength, which is associated with lower back pain [101]. Mannion et al. [77] and Coorevits et al. [78] showed that the test endurance time was highly associated with isometric/endurance hip and back musculature strength (r = 0.84–98, *p* < 0.01). On the other hand, Kankaanpää et al. [79], found that this association was moderate (r = 0.60–0.71, *p* < 0.05). However, when BMI (r= −0.49–0.51, *p* < 0.001) in women and age (r = 0.25–0.29, *p* < 0.05) in men were accounted for in the prediction model, the explained variance increased considerably. Thus, the Biering–Sørensen test might be considered as valid for measuring back muscle endurance strength.

Assessing abdominal muscle functionality is clinically relevant since it is considered to be related to lower back pain [102,103]. The curl-up test, or its different versions, was the field test originally used to assess this capacity. In the present review, no original studies evaluating the criterion-related validity of this test were classified as high quality. An alternative of the curl-up test could be the prone bridging test, an isometric holding test in prone position which is currently being used to supposedly measure abdominal endurance strength. The prone bridging test time is inversely associated with lower back pain [104,105]. In relation to the validity of this test, De Blaiser et al. [80] found a higher activation of the abdominal core musculature during the test than for the back and hip musculature, showing a high association between test time and abdominal endurance strength. Future high-quality studies are necessary to clarify the validity of this test.

It should be noted that no study that analysed the criterion-related validity of lower and upper body endurance strength tests were classified as high quality.

#### 4.2.3. Explosive Strength

The standing long jump is proposed in health-related fitness test batteries in preschool children [106], as well as children and adolescents [107] to assess lower body explosive strength, given its criterion-related and predictive validity. However, to our knowledge, the criterion-related validity of this test has not been studied. Bui et al. observed that the Sargent jump test [81] is not appropriate to evaluate lower body explosive strength, because its overestimates the height of a vertical jump and its accuracy is reduced as the jump height increases (mean difference: 4.4 ± 5.1, *p* < 0.001). Due to the close relationship that lower body maximal/explosive strength has on adult health [22,23], more high-quality studies are required to analyse the criterion-related validity of these tests in future research.

#### 4.2.4. Levels of Evidence

Strong evidence indicated that (a) the handgrip strength test with the elbow extended and with the grip span adapted to the hand size and sex (using the TKK dynamometer) is a valid test for assessing hand maximal isometric strength; and (b) the Biering–Sørensen test offers a valid test for assessing endurance strength of hip and back muscles. Moderate evidence indicated that handgrip strength (Jamar) has acceptable validity for assessing hand maximal isometric strength. Due to (a) the low number of high-quality studies, limited evidence (only one study) was found supporting the use of prone bridging for assessing abdominal endurance strength and the Sargent jump test for assessing lower body explosive strength; (b) the inconsistent results found in multiple high-quality studies, limited evidence was found for the validity of using handgrip strength (DynEx) for assessing hand maximal isometric strength; and (c) the consistent results found in multiple low or very low-quality studies, the curl-up test, or its different versions, are not valid for assessing abdominal endurance strength.

### 4.3. Flexibility

Radiography seems to be the best criterion measurement of flexibility, but goniometry is also used as a criterion measure [108,109].

Goniometers are relatively easy to obtain; nevertheless, their use requires a certain technical qualification since it is a sensitive method, and thus it is not feasible for use in all settings [110]. Traditionally, the sit-and-reach test, originally designed by Wells and Dillon [111], and its different versions, are included in the fitness test batteries for measuring hamstring and lower back flexibility, which are probably the most widely used measures of flexibility [27].

Mayorga et al. [31] performed a meta-analysis to analyse the criterion-related validity of the sit-and-reach and its different versions (modified sit-and-reach, back-saver sit-and-reach, modified back-saver sit-and-reach, V sit-and-reach, modification V sit-and-reach, unilateral sit-and-reach and chair sit-and-reach). These tests showed moderate validity for estimating hamstring extensibility, but low validity for estimating lumbar extensibility. They also found that the classic sit-and-reach test had the highest criterion-related validity coefficient in both hamstring and lumbar extensibility, compared to the other test, which does not seem to justify the use of the classic protocol modifications in order to solve the problems attributed to itself (i.e., the length proportion between the upper and lower limbs or the position of the head and ankles).

The toe-touch test is another field-based test for measuring hamstring flexibility, in which the individuals were assessed standing instead of sitting on the floor [112]. Although this test is easy to administer and can be an alternative to the sit-and-reach test, when the participant has problems being measured sitting, it is not proposed for any filed-based fitness test battery. A meta-analysis [29] analysed the criterion-related validity of the toe-touch test for measuring hamstring flexibility, reporting similar validity coefficients to those of the classic sit-and-reach.

It is interesting to highlight that Nuzzo [113] has recently suggested that flexibility should be invalidated as a major component of fitness, due to its lack of predictive and concurrent validity in terms of meaningful health and performance outcomes.

#### Levels of Evidence

Strong evidence indicated that (a) the sit-and-reach test and its modified versions have moderate validity for estimating hamstring extensibility, but low validity for estimating lumbar extensibility; and (b) the toe-to-touch test has moderate validity for estimating hamstring extensibility. 

### 4.4. Motor Fitness

The validity of motor fitness tests is the least studied in adults. None of the three studies that analysed the criterion-related validity in motor fitness tests were classified as high quality. Given that the motor fitness tests (i.e., gait/walking speed, balance, timed up and go) are associated with all-cause mortality [114,115,116], falls and fractures [117], disability in activities of daily living [118] and depression [119], it would be useful to know their criterion-related validity.

#### Levels of Evidence

Due to the consistent results found in multiple low-quality studies, we found limited evidence that the ten-step test had moderate validity in assessing agility.

## 5. Conclusions

The systematic review emphasized important major points regarding the criterion-related validity of adult field-based fitness tests (Figure 2):

Cardiorespiratory fitness: the 20 m shuttle run tests best assessed cardiorespiratory fitness using Leger’s equation. Alternatively, the 1.5-mile, 12 min run/walk and YMCA step tests were other cardiorespiratory testing options. When low-level cardiorespiratory fitness existed, or if running was possible, the 2 km, then Oja’s equation or 6 min walk tests were appropriate alternatives.

Muscular strength: strong evidence indicated that (a) the handgrip strength test, with the elbow extended and with the grip span adapted to the individual’s hand size (using the TKK dynamometer), offers a valid means to assess hand maximal isometric strength; and (b) the Biering–Sørensen test estimated the endurance strength of hip and back muscles. Limited evidence (only one study) supported the prone bridging and Sargent jump tests as abdominal endurance strength and lower body explosive strength surrogate markers, respectively. 

Flexibility: strong evidence supported the sit-and-reach test and its different versions, and that the toe-to-touch tests is not valid for assessing hamstring and lower back flexibility.

Motor fitness: limited evidence about the criterion-related validity of motor fitness existed. 

When there are problems of space and time, as in clinical settings, the YMCA step and the handgrip strength tests are good alternatives for assessing cardiorespiratory fitness and isometric muscular strength, respectively.

## Figures and Tables

**Figure 1 jcm-10-03743-f001:**
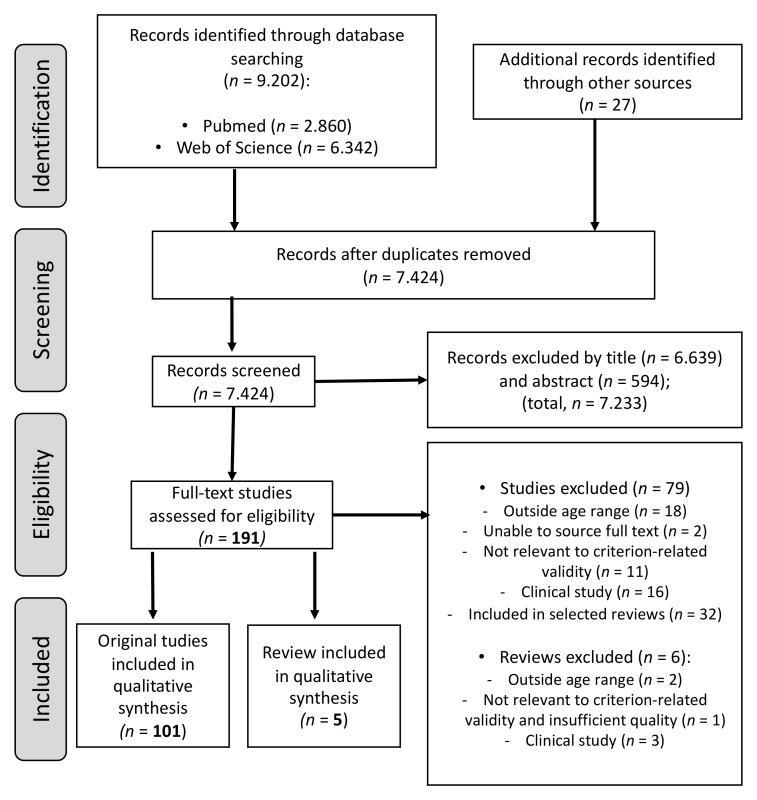
Flow chart of retrieved and selected articles.

**Figure 2 jcm-10-03743-f002:**
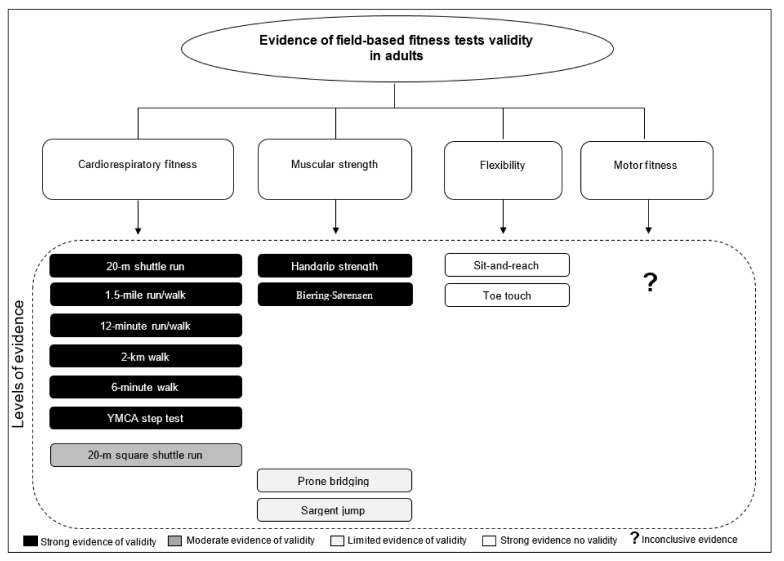
Major points regarding criterion-related validity of adult field-based fitness tests.

**Table 1 jcm-10-03743-t001:** Levels of evidence of cardiorespiratory fitness tests.

Field-Based Fitness Test	Strong	Moderate	Limited
*Shuttle run tests*			
20 m shuttle run			
20 m square shuttle			
Incremental shuttle walk			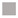
*Distance and time-based run/walk test*			
1.5-mile run/walk			
12 min run/walk			
5000 m run/walk	◐		
3 miles run/walk	◐		
2 miles run/walk	◐		
3.000 m run/walk	◐		
1000 m run/walk	◐		
600 m run/walk	◐		
600 yd run/walk	◐		
½-mile run/walk	◐		
¼-mile run/walk	◐		
9 min run/walk			
2 km walk			
6 min walk			
1-mile walk			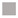
¼-mile walk			
3 min walk			
Treadmill jogging			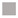
Mankato submaximal exercise			
Modified Astrand–Ryhming			
University Montreal			
Ruffier			ⵔ
*Step tests*			
YMCA step			
Chester step			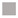
Modified Harvard step			
6 min single 15 cm-step			
Modified Canadian aerobic fitness step			
Tecumseh step			
Astrand–Ryhming step			
Danish step			ⵔ
Queen’s College step			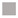
2 min step			ⵔ


 Indicates high validity; ⵔ moderate validity; ◐ low/null validity; 
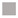
 inconclusive validity.

**Table 2 jcm-10-03743-t002:** Levels of evidence of muscular strength, flexibility and motor fitness tests.

Field-Based Fitness Test	Strong	Moderate	Limited
*Maximal isometric strength*			
Handgrip strength (TKK)			
Handgrip strength (Jamar)		◐	
Handgrip strength (DynEx)			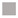
*Hip and back endurance strength*			
Biering–Sørensen			
*Abdominal endurance strength*			
Prone bridging			
Original/modifications curl-up			ⵔ
*Lower body endurance strength*			
Sit-to-stand			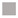
*Lower body explosive strength*			
Sargent jump			
*Upper body endurance strength*			
Original/modification flexed-arm hang			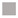
Baumgartner modified pull-up			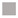
Standard push-up			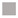
Hand-release push-up			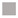
Bent-knee push-up			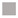
Revised push-up			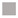
*Lower back flexibility*			
Original/modifications sit-and-reach			
*Hamstring flexibility*	ⵔ		
Original/modifications sit-and-reach	◐		
Toe-touch	◐		
*Agility*			
Ten-step			◐
*Balance*			
Romberg test			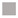


 Indicates high validity; ⵔ moderate validity; ◐ low/null validity; 
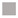
 inconclusive validity.

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
