# Peer review of "Criterion-Related Validity of Field-Based Fitness Tests in Adults: A Systematic Review"

_jcm, 2021, doi:10.3390/jcm10163743_

Round 1
Reviewer 1 Report
thank you for your important SR regarding field based fitness testing for adults. Your methodology appears solid however the writing is awkward and limits clear understanding of your paper. Strongly suggest you enlist the assistance of a professional practitioner with English as a first language to join you on this paper. Please see comments in attached document for parts of your paper. Too time intensive to review the manuscript in its current form.

Author Response
We would like to gratefully thank the Reviewer for his thoughtful and constructive comments, which have undoubtedly improved the quality of our manuscript. We have carefully considered all of the suggestions, and have integrated them into the revised manuscript. Changes to the original manuscript have been incorporated by using yellow background. We believe our manuscript is now stronger as a result of these modifications.
"Please see the attachment"

Reviewer 2 Report
This review provides important information for exercise scientists who are concerned about the validity of commonly used test to assess physical fitness in a broad range of populations. It is very useful and practical.
Just one minor correction:
Line 58: Change “filed-test” to field-test
Author Response
Dear Review,
Please, find enclosed the revised version of our manuscript entitled, “Criterion-related validity of field-based fitness tests in adults: A systematic review" by Dr. Castro-Piñero et al., to be considered for publication in Journal of Clinical Medicine. We would like to gratefully thank the Reviewer for his/her time.
Changes to the original manuscript have been incorporated by using yellow background. An itemized point-by-point response to the Reviewer’ comments is presented below.
"Please see the attachment".
